

# Overland movement in African clawed frogs (*Xenopus laevis*): empirical dispersal data from within their native range

F. André De Villiers and John Measey

Centre for Invasion Biology, Department of Botany & Zoology, Stellenbosch University, Stellenbosch, South Africa

## ABSTRACT

Dispersal forms are an important component of the ecology of many animals, and reach particular importance for predicting ranges of invasive species. African clawed frogs (*Xenopus laevis*) move overland between water bodies, but all empirical studies are from invasive populations with none from their native southern Africa. Here we report on incidents of overland movement found through a capture-recapture study carried out over a three year period in Overstrand, South Africa. The maximum distance moved was 2.4 km with most of the 91 animals, representing 5% of the population, moving ~150 m. We found no differences in distances moved by males and females, despite the former being smaller. Fewer males moved overland, but this was no different from the sex bias found in the population. In laboratory performance trials, we found that males outperformed females, in both distance moved and time to exhaustion, when corrected for size. Overland movement occurred throughout the year, but reached peaks in spring and early summer when temporary water bodies were drying. Despite permanent impoundments being located within the study area, we found no evidence for migrations of animals between temporary and permanent water bodies. Our study provides the first dispersal kernel for *X. laevis* and suggests that it is similar to many non-pipid anurans with respect to dispersal.

## INTRODUCTION

The ability to disperse is present in most organisms (*Clobert et al., 2009*) and is one of their most important characteristics (*Bonte & Dahirel, 2017*). Dispersal entails the individual movement between habitat patches, and as such not only affects individual traits, but also population characteristics, such as community structure (*Bowler & Benton, 2005*; *Doebeli, 1995*; *Holt, 1985*; *Matthysen, 2005*). Dispersal differs between as well as within species (*Altwegg, Ringsby & Sæther, 2000*; *Bowler & Benton, 2009*; *Schneider, Dover & Fry, 2003*; *Stevens, Pavoine & Baguette, 2010*); this is because factors, which include mate finding, habitat quality and competition (both intra and interspecific) influence the costs and benefits relationship of dispersal (*Bowler & Benton, 2009*; *Clobert et al., 2009*). Characterising dispersal differences is increasing its significance as models and simulations require accurate character traits. To characterise the statistical distribution

Corresponding author
John Measey, john@measey.com

of dispersal distances for a species, it is common to produce a dispersal kernel which is a probability density function of the distribution of different Euclidian distances from their source to post-dispersal points (*Nathan et al., 2012*). This is particularly important for invasive species (*Travis et al., 2009*) where dispersal is a key characteristic. The use of dispersal kernels in models of invasion help inform managers of potential invasive spread (e.g., *Vimercati et al., 2017b*; *Vimercati et al., 2017a*).

Animal movement, including dispersal, has been linked to an individual's morphology by studies of laboratory performance (*Arnold, 1983*; *Huey & Stevenson, 1979*). Performance ability represents the animal's maximum exertion, whereas dispersal represents the observed distance moved. As such, performance measures such as endurance may be a good indicator of the relative dispersal ability of that animal. For example, *Herrel, Vasilopoulou-Kampitsi & Bonneaud (2014)* demonstrated that the influence of morphology on jumping performance was sex-specific for *Xenopus tropicalis*, suggesting that this differentially influenced their dispersal ability (see also *Herrel & Bonneaud, 2012*). Morphology ultimately determines the performance ability of the animal and consequently the animal's dispersal ability, which is why natural selection acts upon morphology (*Hertz, Huey & Garland Jr, 1988*; *Zug, 1972*) Another interesting example was observed in cane toads (*Rhinella marina*) where populations at the invasion front were found to have increased dispersal abilities (*Alford et al., 2009*) due to an increase in endurance (*Llewelyn et al., 2010*) resulting from longer leg length (*Phillips et al., 2006*), and shifts in behavioural traits (*Gruber et al., 2017*). This spatial sorting of a population has now been found in an increasing number of invasive species, including the pipid frog, *X laevis* (*Courant et al., 2017a*; *Louppe, Courant & Herrel, 2017*).

*Xenopus laevis* occurs throughout southern Africa, occupying almost every aquatic habitat found within this range (*Furman et al., 2015*; *Measey, 2004*). The frogs in this genus are highly adapted for an aquatic lifestyle (*Trueb, 1996*), as most of their life is spent in the water. It has been suggested that dispersal is facilitated through aquatic corridors (i.e., rivers, streams, irrigation ditches, etc.) leading to the classical view that these frogs are fully aquatic (*Fouquet & Measey, 2006*; *Lobos & Measey, 2002*; *Measey & Channing, 2003*; *Van Dijk, 1977*). However this view has been challenged by many observations of overland movements (reviewed by *Measey, 2016*), which suggest that these frogs are capable of dispersing overland, and that they might better be termed "principally aquatic". However, the majority of the literature represents anecdotal or inferred movements with little or no information on what proportion of the population disperses, what time of year they disperse and the function of the dispersal kernel. It has been suggested that *X. laevis* migrates (*Hey, 1949*), adult movements between permanent and temporary habitats, but no data exist to substantiate this (*Measey, 2016*). It is important to clarify whether or not individuals migrate back to ponds or disperse between ponds, as regular dispersal would mean that ponds do not represent discrete populations and instead that *X. laevis* may meet some of the conditions for a metapopulation (cf *Smith & Green, 2005*).

Literature on the ecology of *X. laevis* is growing rapidly, due to increasing numbers of studies of invasive populations (e.g., *Amaral & Rebelo, 2012*; *Courant et al., 2017b*; *Lillo, Faraone & Valvo, 2011*). However, there have been very few empirical studies conducted within its native range, despite the species being almost ubiquitous in southern Africa. Here

*X. laevis* is associated with artificial impoundments, such as farm dams, sewage works, fish hatcheries, etc. (*Schoonbee, Prinsloo & Nxiweni, 1992*; *Van Dijk, 1977*) while their presence in natural water bodies goes almost unnoticed Intriguingly, this species appears to occur in impoundments in desert areas where it is likely to have had anything but a transient presence, making it the most widespread amphibian species in South Africa (*Measey, 2004*). This might be because there is a tendency for human mediated dispersal of *X. laevis* for fishing bait and via universities (*Measey et al., 2017*; *Van Sittert & Measey, 2016*; *Weldon, De Villiers & Du Preez, 2007*) Poynton (quoted in *De Moor & Bruton, 1988*) was of the opinion that *X. laevis* made use of artificial water bodies to expand their range and become an extra-limital species, and it has been suggested that there is leading edge dispersal in *X. laevis* which could explain the near ubiquitous distribution (*Measey et al., 2017*; *Van Dijk, 1977*). However, the extent to which populations disperse between natural and artificial impoundments in their native southern Africa is largely unknown (*Measey, 2016*).

To redress the dearth of data on overland movement from native populations, we conducted a capture-mark-recapture exercise with *X. laevis* in eight water bodies in the Overstrand region, southwestern South Africa. We conducted a study over 33 months asking the same four questions posed by *Measey (2016)* of our data: (1) Is there evidence for overland dispersal in a native population of *X. laevis*; (2) What distances are moved overland; (3) Is there evidence that overland movement is seasonal or associated with rain or drying habitats; (4) Is there evidence of overland movement being migratory with respect to breeding? Lastly, we use a small laboratory study of maximum performance of endurance to ask whether sex and size differences match population movements in the field.

## MATERIALS & METHODS

### Study site

The study area (34.325°S, 19.103°E) 8 km east of Kleinmond (hereafter referred to as Kleinmond) covers an area of 5.4 km$^2$ (3.5 km by 1.5 km) in the Overstrand, Western Cape Province, South Africa (Fig. 1). It falls within a single catchment and is relatively homogenous with a very gentle slope running approximately north–south with a change of less than 10 m altitude in 1.5 km. The eight water bodies are divided between five temporary waterbodies (vleis: typically full between July and November) and three permanent impoundments (dams) which contain water all year, but may vary in depth. In each case the permanent impoundments are artificial, while the vleis are natural. The study area had a maximum distance between water bodies almost spanning the entire area (3.7 km), while the minimum distance was 91 m. Three streams run north-south through the study area, but flow only part of the year. A paved road and several unpaved roads run through the area (Fig. 1)

*Xenopus laevis* is known to occur in all water bodies, but the congeneric *X. gilli* only occurs in vleis (see *De Villiers, De Kock & Measey, 2016*; *Fogell, Tolley & Measey, 2013*; *Furman et al., 2017*; *Vogt et al., 2017*). The dams have only *X. laevis*, *A. fuscigula* and *S. capensis*. Vegetation is lowland sand-stone fynbos (*Mucina & Rutherford, 2006*), with

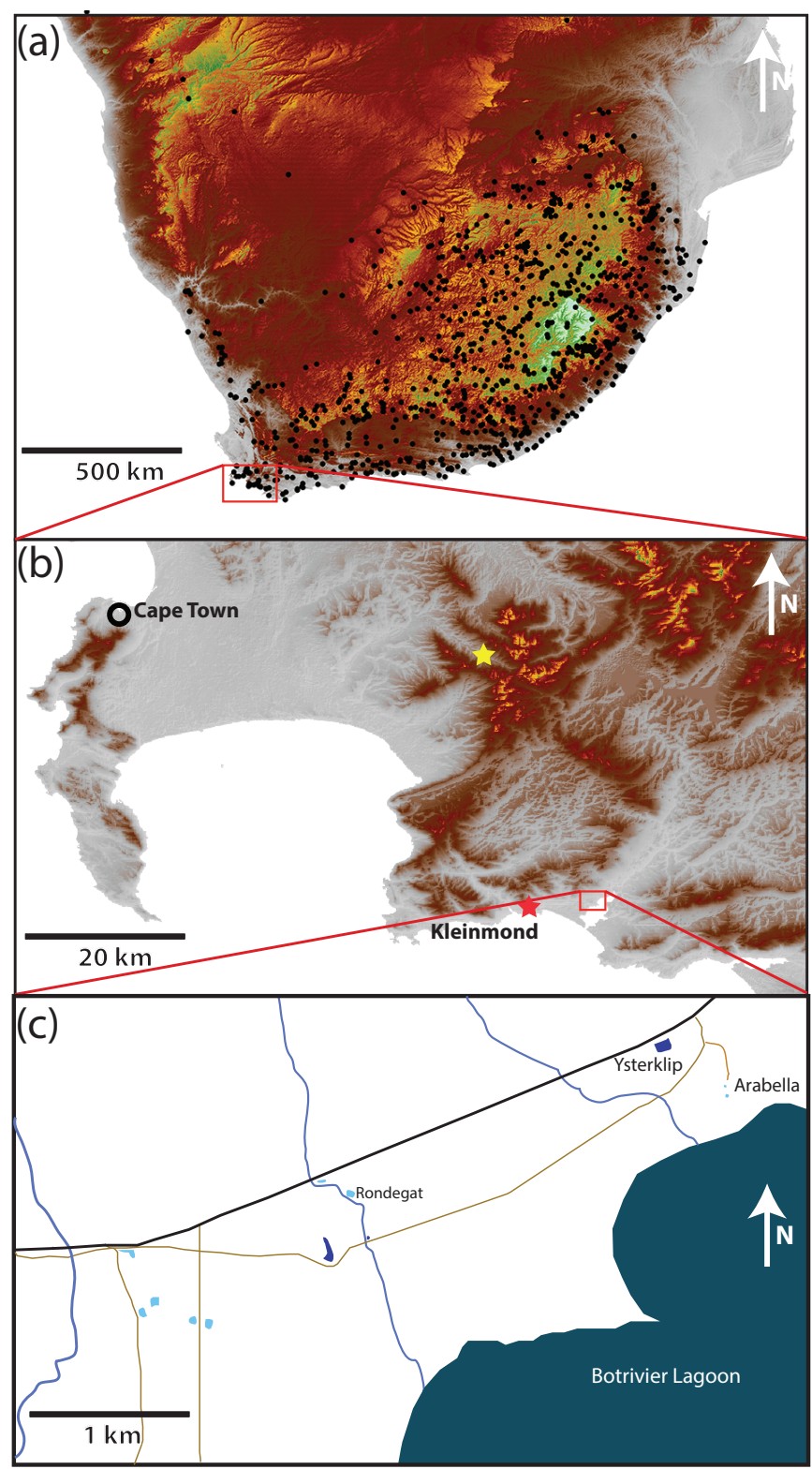

**Figure 1  Position of the study site.** (A) Southern Africa, with black dots showing known locations of *Xenopus laevis*. The data demonstrates the widespread distribution over (continued on next page...)

**Figure 1 (…continued)**
the full altitudinal range of the region from the highlands of Lesotho (>3,000 m asl in white), through the Highveld of the escarpment (1,000 to 2,000 m asl in brown and yellow, respectively) to the lowland seaboard in grey (<500 m asl). (B) The site in the extreme southwest is 8 km East of Kleinmond (red star) borders the Botrivier Estuary to the South. The position of Jonkershoek (45 km north–west of the study site) is shown with a yellow star. The area is chosen to show the Cape Fold Mountains (brown to yellow) and extensive lowland (grey) areas.

areas particularly heavily invaded by *Acacia saligna* (Port Jackson Willow), *Hakea sericea* (Silky Hakia) and *Acacia mearnsii* (Black Wattle).

It is noteworthy that the southwestern Cape of South Africa was undergoing a drought at the time of this study and periodicity of temporary water was affected. The temporary pools held water for six and seven months in the first and second years of the study, and these times did not coincide with seasonal changes due to a lag between the onset of rains and the filling (and emptying) of pools. In contrast the permanent dams contained water throughout the study, although the levels changed considerably.

## Capture-mark-recapture

Frogs were collected from January 2014 to June 2016 by using baited traps (bucket or fyke traps; see *Lobos & Measey, 2002*; *Vogt et al., 2017*). Three to five traps were set in five temporary and three permanent ponds (Fig. 1). Each trapping session was conducted for all water bodies for which water was deep enough to contain a trap (>20 cm deep), from one to four consecutive nights, each night the traps were set and collected again the following morning. Regular trapping with three to four week intervals between sessions took place from 2014 to the end of 2015. Thereafter trapping has taken place at six-monthly intervals. All animals were processed at the edge of each site and returned to the site in which they were trapped. In addition to the eight regular trapping sites, in June 2016 we placed traps for three nights in three water bodies immediately outside of the area, but none were found to contain *X. laevis*, leading us to believe that we were covering a discrete population. Ethical clearance for capture-mark-recapture was obtained from Stellenbosch University (SU-ACUD14-00028) and permits were issued from CapeNature (AAA007-00092-0056).

Upon emptying traps, all frogs were first scanned with a hand-held scanner (APR 350, Agrident, Barsinghausen Germany) and the unique number recorded together with the locality of the individual. Frogs (>30 mm SVL) without tags were then tagged using 8 mm PIT tags, which are small glass capsules with an electromagnetic coil (*Guimaraes et al., 2014*). The tag was placed in 15-gauge hypodermic needle and injected underneath the skin above the dorsal lymph sac (*Donnelly et al., 1994*). Each frog was photographed dorsally on a 10 × 10 mm grid. Image numbers were recorded together with tag numbers, and the scaled images used to calculate (SVL) using ImageJ (*Rasband, 2012*). Frogs were sexed externally by the presence of labial lobes in females and nuptial pads on the forearms of males (see *Measey, 2001*). Generally, it was possible to sex individuals greater than 45 mm snout-vent length (SVL), and smaller animals were classified as juveniles if sex could not be unambiguously determined.

## Performance measures

Twenty (10 male and 10 female) *X. laevis* were collected from Kleinmond, and transported to and housed at Stellenbosch University's Department of Botany and Zoology. Each frog was PIT tagged and housed in its own aquarium at a constant temperature of 20 °C (*Careau et al., 2014*; *Herrel et al., 2012*; *Louppe, Courant & Herrel, 2017*). Animals were fed every second day with sheep's heart, *ad libitum*, and weighed once a week to monitor their well-being Ethical clearance for performance measures was obtained from Stellenbosch University (SU-ACUD14-00028).

Prior to performance trials, each animal was measured using digital callipers (to the nearest 0.01 mm). Measurements were taken as follows: body length (SVL: the length from the tip of the snout to the cloaca (*Herrel et al., 2012*), the length and width of the head, the length of the jaw radius, humerus, hand, longest finger, longest toe foot tibia femur length and width of the ilium and interaxial distance (i.e., a lateral measurement of the vertebral and ilium length: *Herrel et al., 2012*; *Louppe, Courant & Herrel, 2017*) where appropriate, all measurements were size (SVL) corrected for comparison.

All performance trials were conducted within three weeks of capture in a controlled environment with a constant temperature of 20 °C ($\pm$2°), as this is the optimal performance temperature for *X. laevis* (*Miller, 1982*). All animals were rested for at least 24 h between trials, with each animal undergoing three trials where the longest distance in the shortest time was retained for analysis. Dry endurance was determined on a 4 m circular track with a rubber grip mat as substrate. Each trial was timed and the distance moved was calculated from the number of laps with continuous movement insured by tapping the frog between the hind legs. The trial was considered finished if the frog refused to move after multiple taps, and was unable to right itself (*Herrel & Bonneaud, 2012*).

## Data analysis

In order to assess potential bias in capture rates in our dataset, we first compared sex ratios and sizes of animals that were captured once (26.7%) with those that were captured more than once. A chi-squared test ($\chi^2$) showed that sex ratios were the same for animals that were captured once or more than once ($\chi^2 = 0.012$, *p*-value $= 0.9123$), but an ANOVA showed that there was a significant difference in size (see below). Therefore to test for sex bias in dispersal, we use the entire dataset. But for size, we use only those animals which were captured more than once. In each case, a $\chi^2$ test was used in R (*R Core Team, 2017*) with a *P*-value based on 10,000,000 bootstraps.

Sex ratio was calculated as the number of males per 100 females per capture session. The distance between the pond of origin (i.e., the pond where the frog was tagged) and the destination pond were measured (to the closest meter) using ArcGIS (Version 10.2; ESRI, Redlands, CA, USA). As such this represented the Euclidian distances between sites. Dispersal distances were log transformed to meet assumptions of homoscedasticity. Normality of data was determined by using QQ-plots and the homogeneity of the variances were determined by using Levene's test. Occurrence of movements (where an animal was tagged in one location, but recaptured in another) were coded according to whether or not they occurred within one season (dry: December to May; wet: June to November), sex

and the size of the individual at first capture. The dataset for comparison was made up of individuals that were marked and recaptured during the same period within one of the ponds.

The dispersal kernel was fitted using all dispersal distances (including instances where individuals moved more than once). We used the fitdistrplus package (*Delignette-Muller & Dutang, 2015*) in R v3.3.3 (*R Core Team, 2017*) to test the fit of the data against four distribution types: exponential, lognormal, Weibull and gamma. We then inferred the best fit through minimum AIC. All means are reported ± Standard Deviation.

For the performance data, we logged all linear measurement data to fulfill assumptions of normality and homoscedasticity. To test for differences between sexes, we conducted a MANCOVA (*R Core Team, 2017*) with all morphological measurements as dependent variables, with size (SVL) as a covariate with Wilk's Lambda statistic. Next, we conducted another MANCOVA using the (log of) maximum distance moved and time to exhaustion as dependent variables, with size (SVL) as a covariate.

## RESULTS

### Capture-mark-recapture

We made 9,401 captures of 1,755 individual *X. laevis* in 80 capture nights over 28 sessions in 3 years. The mean number of animals captured per session was 354 (±258.2: range 1,128 to 15), and the sex ratio was always female biased, varying from 30 to 74 males per 100 females. Animals found to have been captured more than once were larger ($n = 1,303$; SVL 64.95 ± 14.96 mm) than those not recaptured ($n = 452$; SVL 61.39 ± 13.80 mm; $F_{1,1,753} = 20.609$; $P < 0.0001$), the discrepancy being made by smallest marked individuals (30–49 mm SVL) for which sex could often not be determined ($n = 81$), and which did not move. Juveniles (<45 mm) made up a significant part of some capture sessions, averaging 4.1% (±4.64). We noted large numbers of metamorphs at one of the sites (Ysterklip), but animals <30 mm SVL rarely entered our traps. The majority of individuals was recaptured at least once ($n = 1,303$), with only 34.7% of individuals ($n = 452$) captured only once. We found significant differences between the sizes of males (SVL mean 63.2 ± 9.12 mm; max 93.0 mm; $n = 673$) and females (SVL mean 66.4 ± 16.60 mm; max 129.6 mm; $n = 1001$: $F_{1,1,672} = 9.318$; $P = 0.0023$).

Ninety-one individuals (5.2%) moved between one and four times (mean 1.19 ± 0.576) over the entire period. Of the 11 animals that moved two or more times, five returned to their original site of capture. The modal overland distance moved was 147 m, with the frequency of small movements far exceeding long ones (Fig. 2). A lognormal distribution fitted the highest dispersal values best as well as performing well on the mid-range values. However, all four distributions fitted the data well, differing by less than 60 δAIC values (Table 1). Equation (1), gives the probability density function, where $Y$ is the expected frequency of moving frogs and $x$ is the distance moved

$$Y = \frac{1}{0.62x\sqrt{2\pi}} e^{-\frac{(lnx - 5.575)}{0.765}}. \tag{1}$$

The nature of capture-recapture using our baited traps does not allow for the precise timing of the majority of movements that occurred between capture sites. Of the 69 individuals for

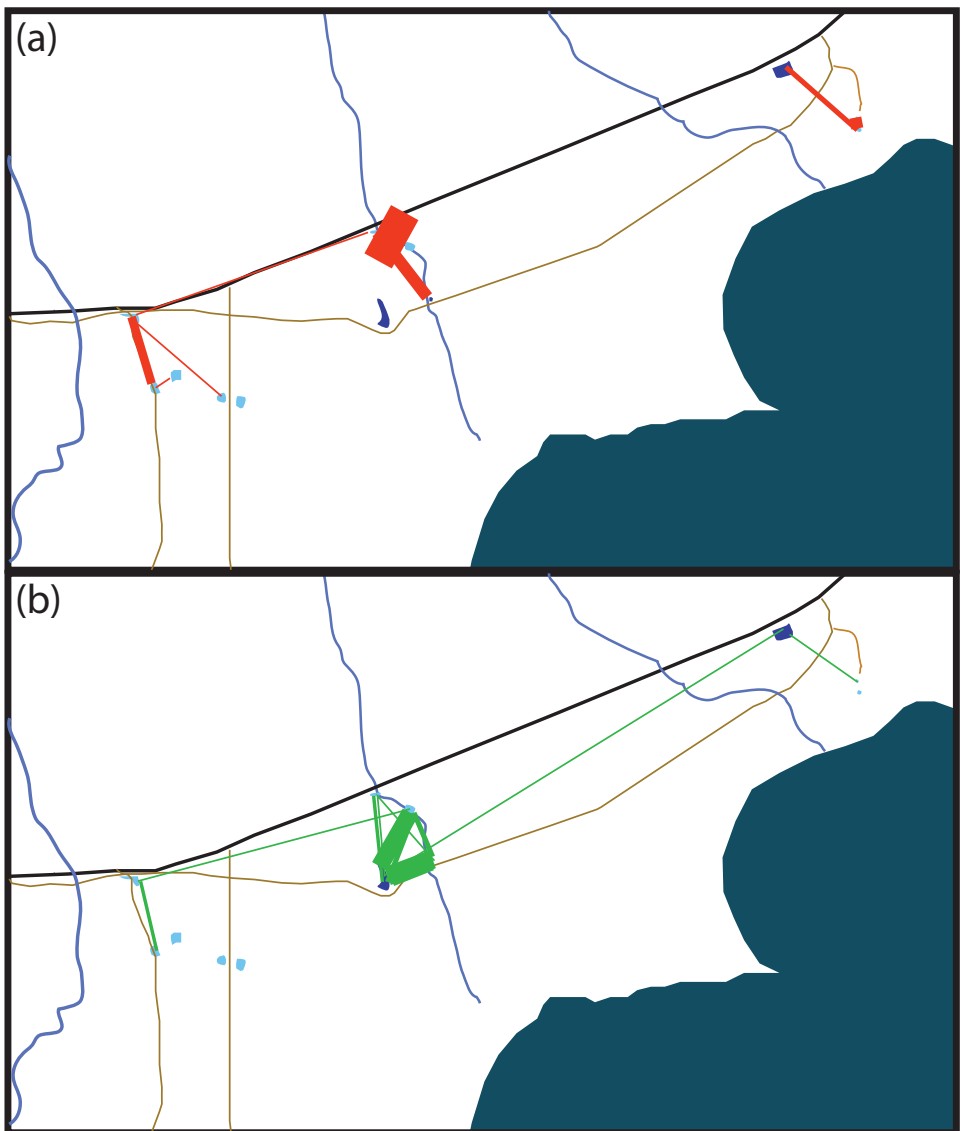

**Figure 2  Schematic of movement by marked *Xenopus laevis* in (A) summer and (B) winter between water bodies.** Thickness of red (A) and green (B) lines is proportionate to the amount of dispersal movements within that season. The site 5 km East of Kleinmond has natural temporary vleis (light blue) and anthropogenic impoundments (dark blue), and lies north of the brackish Botrivier Estuary. Paved (black line) and unpaved (brown lines) roads run through the area together with three temporary streams (blue lines).

which the season of movement was known because they were recorded at both the origin and destination sites in the same season, there was no difference between seasons in either the number of animals moving ($\chi^2 = 0.552$, $p = 0.519$; dry = 57, wet = 48) or in the sex of animals moving ($\chi^2 = 0.552$, $p = 0.519$ dry sex ratio = 39/18 , wet sex ratio = 36/12). However, we did find that individuals moved significantly farther during the wet period (mean = 387.5 m; median = 344.9 m) than during the dry (mean = 245.4 m; median

**Table 1** **Model testing for distributions of dispersal kernel of *Xenopus laevis* from Kleinmond.** Models are based on capture-mark-recapture data of 108 movements of 91 individuals over a three-year period.

| Distribution | Akaike's information criterion (AIC) | ΔAIC | Bayesian information criterion | Anderson–Darling statistic |
|---|---|---|---|---|
| Log normal | 1,410.92 | 0 | 1,416.29 | 3.069861 |
| Gamma | 1,433.32 | 22.40 | 1,438.69 | 3.927477 |
| Weibull | 1,450.97 | 40.05 | 1,456.33 | 5.497646 |
| Exponential | 1,469.55 | 58.63 | 1,472.23 | 10.39505 |

$= 147.7$ m; $F_{1,101} = 6.833$, $p = 0.0103$). The maximum dispersal distance observed was a *X. laevis* female, which travelled 2.42 km in less than six weeks, while another *X. laevis* female dispersed 1.36 km in less than three weeks. Neither of these movements was downhill or appeared to follow any form of stream or movement of water overland. In addition, we have a record of a single male animal that was caught in one temporary water site on one night and in another 147 m away on the next night. We also found some synchronous movements. For example, in October 2015 we captured five animals in Arabella that were all captured two nights later 91 m away, and in February 2015 we captured six animals at Rondegat that were all captured two months later in another water body 147 m away.

Many of the movements during the dry summer season happened between adjacent temporary sites (Fig. 3A). However, during the wet winter season movements happened between temporary and permanent sites, as well as between temporary sites (Fig. 3B). The timing of these movement events was related to drying of temporary water sites in December 2014 and in October 2015 which also coincided with drying events after very poor winter rains (Fig. 4). These two events encompassed the majority of movement events (58.6%), but we recorded movements during almost every capture session (83%; Fig. 4).

Even though we had twice the number of females ($n = 63$) moving as males ($n = 28$), this was not significantly different from the sex ratio of animals that did not move (females 988; males 672; $\chi^2 = 3.668$, $P = 0.0630$). Females (mean $= 330.0$ m; median $= 249$ m) moved no farther than males (mean $= 324.0$ m; median $= 249$ m; $F_{1,101} = 0.002$, $p = 0.967$). No significant bias was found in the size of animals that were moving ($n = 91$; mean $= 66.5 \pm 15.47$ mm; median $= 63.4$ mm) compared to those that were recaptured only within the same water body ($n = 1\,213$; mean $= 64.8 \pm 14.94$ mm; median $= 63.4$ mm; $F_{1,1,302} = 1.200$, $p = 0.274$).

## Performance

We found significant differences between the sizes of male and female *X. laevis* within the small subset ($n = 20$) which we tested for performance ($F_{1,18} = 10.4$; $P = 0.0047$). In this subsample, the size-corrected forelimb measures of males were significantly longer than those of females. However, the relative length of the longest toe on the foot was longer in females (Table 2). We found a difference in the size-corrected distance moved by the two sexes before exhaustion, with males moving significantly farther (Table 2). Time to exhaustion, when both sex and corrected size were included, was significant (Table 2), with
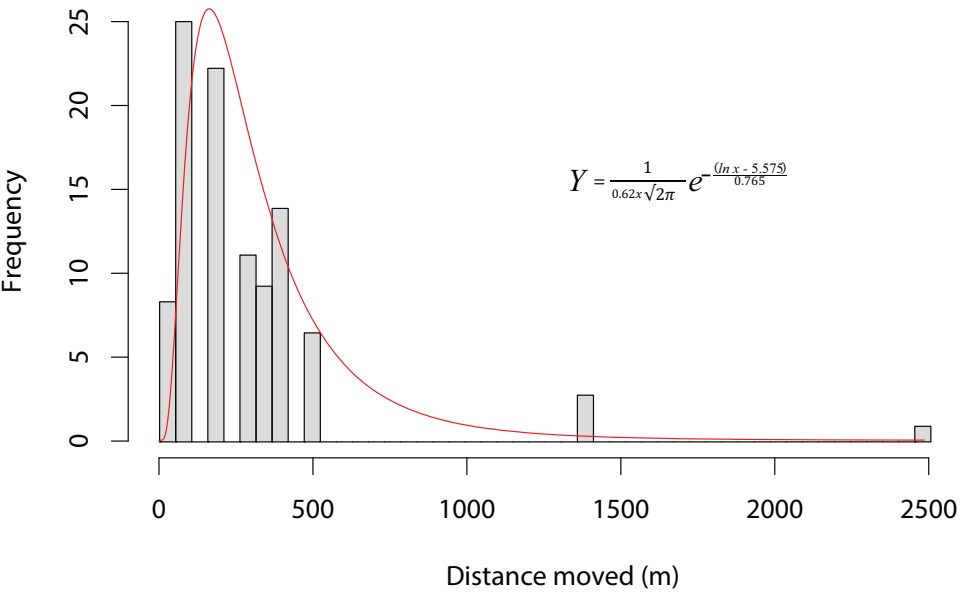

**Figure 3** **Dispersal kernel of *Xenopus laevis* at a site near Kleinmond, South Africa.** Bars show the frequency of distances moved between water bodies by individuals during capture-mark-recapture based on 108 movements of 91 individuals over three years. The data is best described by a lognormal curve (red line), for which the probability density function is provided ($Y$ is the expected frequency of moving frogs, and $x$ is the distance moved).

smaller males moving for a longer time than larger females. The mean distance moved was 24.9 m ($\pm$3.53 m) in around 3 min (187 $\pm$ 36.46 s).

## DISCUSSION

We present the first empirical data for overland movement of *X. laevis* within its native range, demonstrating that distances moved are up to 2.42 km over a period of less than 6 weeks. This finding is an important extension to the data reviewed by *Measey (2016)* in which the longest distance moved was 2 km in an invasive population. In addition to extending the maximum distance moved overland, we were able to calculate a dispersal kernel for this species. Over a period of three years, we found that 5% of individuals moved between sites, although this does not necessarily mean that 95% of the animals were philopatric, as we captured 26% of animals only once.

### Do *Xenopus laevis* migrate?

*Hey (1949)* provided a description of *X. laevis* in Jonkershoek (45 km north–west of our study site) involved in a migration from permanent impoundments into freshly filled vleis in order to breed. We had expected that the combination of permanent and temporary water bodies in our study area would allow us to collect data on such migration events over the three years of study, but we found none. Only five animals were found to return to their original site of capture, but these movements were not necessarily through permanent waters. Instead, we presume that animals that left the temporary water went
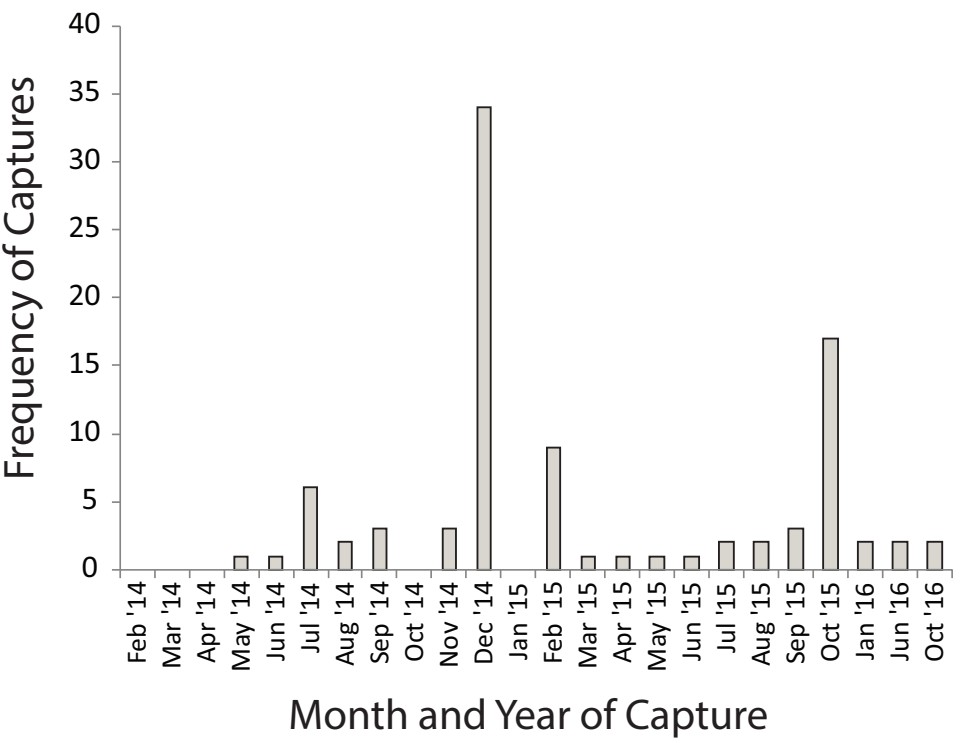

**Figure 4 Numbers of *Xenopus laevis* caught per month in a pond other than the one in which they were marked.** Spikes in December 2014 and October 2016 coincide with the drying of temporary water-bodies in those years.

into subterranean aestivation (cf *Balinsky et al., 1967*), although efforts to find any *Xenopus* through excavations in the area proved unsuccessful (J Measey, 2015, unpublished data). Whether these animals hide collectively or are scattered throughout the area is potentially important. Attempts at eradicating invasive populations may flounder if a proportion of buried animals goes undetected. Like *Measey (2016)*, we cannot discount the possibility that *X. laevis* do migrate between water bodies under particular circumstances, but we found no evidence of this in the Kleinmond population. However, we did find examples of synchronous movements, both during the wet and dry periods. Movements of large numbers of *X. laevis* have been witnessed (*Lobos & Jaksic, 2005*; *Measey, 2016*), but our data suggest that this may also happen on a smaller scale.

### Sexual or size difference in dispersal?

Our study found no bias in sex of animals dispersing, once the highly skewed sex ratio was considered. Other studies have found similar skews toward females (*Lobos & Measey, 2002*). It could be that there is dispersal bias toward smaller life-history stages (metamorphs and juveniles, see *Sinsch, 2014*), although we also found that the smallest animals we tagged did not move. If there were a male-biased dispersal (as might be expected *Hamilton & May, 1977*; *Trochet et al., 2016*), high mortality might help explain the skewed sex ratio. Metamorph survival might be altered in densely populated water bodies such as these

**Table 2  Morphological variables measured for *Xenopus laevis* used in performance tests.** Results from a MANCOVA (with SVL as the covariate) show that sexes are morphologically different. Means (in mm ± Standard Deviation) are given for measures of females ($n = 10$) and males ($n = 10$), together with tests which show that significant differences centred on limb measures. Distance (in m ± SD) and Time (in s ± SD) for stamina performance trials are also given. Stars (*) indicate statistical significance.

| Effect | Variable | Wilk's Lambda | Mean Females | ±SD | Mean Males | ±SD | F | df | P | |
|--------|----------|---------------|--------------|-----|------------|-----|---|----|----|----|
| Sex | | 0.035 | | | | | 17.52 | 11 | 0.0004 | *** |
| | Mass | | 38.46 | 20.524 | 25.60 | 12.777 | 0.23 | 1 | 0.6381 | |
| | Ilium length | | 31.53 | 5.750 | 26.25 | 3.705 | 2.52 | 1 | 0.1306 | |
| | Ilium width | | 14.41 | 2.253 | 13.12 | 1.926 | 1.63 | 1 | 0.2195 | |
| | Femur | | 22.31 | 3.654 | 20.66 | 2.694 | 3.24 | 1 | 0.0898 | |
| | Tibia | | 22.37 | 4.217 | 20.64 | 2.473 | 3.63 | 1 | 0.0738 | |
| | Astragalus | | 15.95 | 3.459 | 14.55 | 2.366 | 3.45 | 1 | 0.0808 | |
| | Longest toe | | 23.89 | 3.858 | 22.87 | 3.256 | 41.59 | 1 | <0.0001 | *** |
| | Humerus | | 11.27 | 1.929 | 11.39 | 1.619 | 10.64 | 1 | 0.0046 | ** |
| | Radius | | 10.27 | 1.854 | 11.55 | 2.028 | 36.07 | 1 | <0.0001 | *** |
| | Hand | | 4.26 | 0.758 | 4.08 | 0.728 | 17.1 | 1 | 0.0007 | ** |
| | Longest finger | | 10.32 | 1.777 | 10.02 | 1.619 | 4.97 | 1 | 0.0395 | * |
| | Distance | | 23.23 | 1.491 | 26.63 | 4.203 | 10.4 | 1 | 0.0047 | ** |
| | Time | | 167.50 | 33.033 | 206.50 | 29.504 | 5.113 | 1 | 0.0182 | * |

(see *De Villiers, De Kock & Measey, 2016*) because adults are cannibalistic (*Measey et al., 2015*; *Vogt et al., 2017*). Capture-mark-recapture studies on adult amphibians generally suggest no sex biased movement (e.g., *Sinsch, 2014*; *Smith & Green, 2006*). In fact, examples of female-biased dispersal (*Austin et al., 2003*; *Lampert et al., 2003*; *Palo et al., 2004*) and male biased dispersal (*Liebgold, Brodie & Cabe, 2011*) in amphibians have been revealed using genetics, which suggest that juveniles (individuals that we cannot tag in most of amphibian species) are responsible for most of dispersal events (inducing gene flow). Because more females than males moved, a genetic study on our population would indicate female-biased dispersal. However, this would not be due to a bias in dispersal of individuals, but simply reflect the already skewed population sex ratio.

The fact that females were found to disperse provides important information for phylogeographic studies using mitochondrial DNA (*De Busschere et al., 2016*; *Furman et al., 2015*; *Measey & Channing, 2003*). African clawed frogs are known to form well defined mtDNA clades in southern Africa, and these have been shown to correspond to sufficiently rapidly evolving nuclear DNA (*Furman et al., 2015*; *Furman et al., 2017*). Presumably, these clades represent areas where both males and females are equally inhibited from dispersing. The extent of dispersal in this study may exceed the first condition for a metapopulation: that ponds sampled would represent local breeding populations (*Hanski, 1999*). A 5% movement of adults between ponds may effectively unite the study site into a single unit, or large patch, with the possibility that a metapopulation occurs at a larger spatial scale (*Smith & Green, 2005*). Existing genetic studies are too coarse to test the potential effective scale at which metapopulations exist within *X. laevis*. More fine scale genetic sampling could

inform the scale at which dispersal occurs as well as what constitutes a dispersal barrier for this species.

In addition to the field data, we present a performance dataset that suggests that males and females are equally able to move long distances. This required males to move proportionately farther and longer than females before exhaustion. Distances moved in our laboratory study are around double those reported by *Louppe, Courant & Herrel (2017)* for two invasive populations of *X. laevis* in France. Similarly, our animals had higher stamina, being able to move for longer before exhaustion. The studies differed in the temperature the trial was conducted (22 °C in France and 20 °C in South Africa). Despite these differences, both studies found that males moved relatively farther for relatively longer such that they were able to perform as well as larger females. The increased performance for males is not explained by their limb morphology, as both studies show males have longer forelimbs (linked with mating), but that hindlimbs are the same length, relative to size, with the exception of longer toes (larger feet) in females. The time taken to move before exhaustion would make it easily possible for animals to move between close sites in a single night as we observed. However, the longest distances observed would likely have taken many nights, and may have included periods in water bodies between sites.

It is noteworthy that within our study site there were two temporary streams (Fig. 2A); one not associated with any water bodies, and the other with three of the sites. While these three sites received the most animals moving between them, this was largely confined to dry periods when the water did not flow. This suggests that *X. laevis* are not reliant on watercourses to guide their movements. However, when the weather is dry watercourses may offer increased levels of humidity that reduce dehydration during overland movements. Dehydration remains an important risk for *X. laevis* moving overland (*Hillman, 1978*), as has been emphasised for other amphibians (*Tingley, Greenlees & Shine, 2012*).

### Seasonality and habitat drying

Our data demonstrate that African clawed frogs move overland throughout the year, and that this behaviour is not restricted to periods of winter rainfall. However, during the wet winter period individuals moved farther. Both observations match the recent literature review (*Measey, 2016*). Additionally, movements between water bodies peaked at the same time that the vleis were drying. This suggests that the majority of animals move some distance in order to aestivate, and do not simply burrow into the mud of a drying pond (although this has been observed, see *Measey, 2016*).

### Conclusion

We found that 5% of *X. laevis* moved between water bodies within an area of 3 km$^2$, with examples of animals moving nearly the full length of the study site. This suggests that frequency of movement within the study site may exceed that required to maintain a metapopulation structure, and instead represent a large patchy population (cf *Smith & Green, 2005*) in what has previously been thought of as a purely aquatic species (see *Measey, 2016*). Although longer distances were moved overland during the wet period, animals moved year round. More females moved than males, but this was in proportion with the

sex bias observed in the population. Males and females moved the same distances between sites, even though males are significantly smaller; identical to results found in previous performance studies (*Louppe, Courant & Herrel, 2017*). Animals found in temporary water bodies did not move into permanent impoundments, despite their presence in the area. We suggest instead that these animals are aestivating underground at an unknown location. This is the first empirical data of overland movement within the native range of *X. laevis*, and the largest mark-recapture study conducted on this species to date.

## ACKNOWLEDGEMENTS

We thank the Michael Lannoo and Audrey Trochet for their constructive reviews, and Donald Kramer for extensive help improving readability of the manuscript. Julien Courant gave comments on previous version of this manuscript. Atherton de Villiers and Andrew Turner (CapeNature) provided institutional and guiding support throughout this project. Many members of the MeaseyLab helped with trapping and the search for buried animals. We would like to extend our special thanks to the landowners and tenants of the study area near Kleinmond.

### Funding

The National Research Foundation (NRF) of South Africa (NRF Grant No. 87759 to John Measey) provided financial support. F. André De Villiers and John Measey received financial and logistical support from the DST-NRF Centre of Excellence for Invasion Biology (CIB). This project received support from the BiodivERsA project (BiodivERsA BR/132/A1/INVAXEN) "Invasive biology of Xenopus laevis in Europe: ecology, impact and predictive models". The funders had no role in study design, data collection and analysis, decision to publish, or preparation of the manuscript.

### Grant Disclosures

The following grant information was disclosed by the authors:
The National Research Foundation: 87759.
DST-NRF Centre of Excellence for Invasion Biology.
BiodivERsA project: BiodivERsA BR/132/A1/INVAXEN.

### Competing Interests

John Measey is an Academic Editor for PeerJ. The authors declare there are no competing interests.

### Author Contributions

- F. André De Villiers conceived and designed the experiments, performed the experiments, reviewed drafts of the paper.
- John Measey conceived and designed the experiments, performed the experiments, analyzed the data, contributed reagents/materials/analysis tools, wrote the paper, prepared figures and/or tables, reviewed drafts of the paper.

## Animal Ethics

The following information was supplied relating to ethical approvals (i.e., approving body and any reference numbers):

Ethical clearance for capture-mark-recapture was obtained from Stellenbosch University (SU-ACUD14-00028).

## Field Study Permissions

The following information was supplied relating to field study approvals (i.e., approving body and any reference numbers):

Field experiments were approved by CapeNature (AAA007-00092-0056).

## Data Availability

The raw data has been provided as a Supplemental File.

## Supplemental Information

Supplemental information for this article can be found online at http://dx.doi.org/10.7717/peerj.4039#supplemental-information.

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
