# Peer review of "Overland movement in African clawed frogs (Xenopus laevis): empirical dispersal data from within their native range"

_PeerJ, doi:10.7717/peerj.4039_

## Round 0.1 · original submission · Minor Revisions

Overview

Both reviewers recommended this manuscript for publication following changes which were primarily issues of clarification of methods and writing. My own reading of the manuscript agrees with their evaluation in general, but I found numerous other specific problems of word use, grammar and punctuation that need attention. In addition, there were several points where conclusions did not seem to be supported by relevant analyses and some potentially useful analyses were missing. Finally, the Supplementary Information seemed incomplete; I did not see data from the non-movers and from the laboratory study.

Reviewer 2 recommended that you consider using all 3 trials for each individual for your performance analysis. Although he makes a valid point that the robustness of the conclusions might be strengthened, I am also aware that your approach is quite common when researchers are attempting to identify maximum capacity and need to eliminate trials that would lower the average due to potentially extraneous factors. I suggest that you think about this and briefly justify whatever approach you take, possibly including references to others taking a similar approach to what you decide.

In my comments below, I raise numerous issues, including some conceptual and statistical ones. Because I am not an expert in amphibian dispersal and my statistical knowledge is fairly limited, I may have failed to understand some aspects of your manuscript. You may treat my comments as a third review, i.e., follow the suggestions where appropriate and provide a detailed rebuttal where the suggestions are not valid. In the attached pdf, I have made numerous grammatical, spelling, and punctuation suggestions, highlighting the questionable section and using inserted comments to provide an alternative. In some cases, I highlighted a section without suggested changes where I was confident that there was no ambiguity about the problem. For some repeated errors, I did not note every occurrence; when you make grammatical and punctuation corrections please search the manuscript to find all needed corrections of the same type.

Editor's Comments

L12. Check spelling of 'water bodies'. I think it is normally two words, unhyphenated, although sometimes written as a single word 'waterbodies'. Correct throughout manuscript if appropriate.
L35. Confusing sentence: it is not very clear how dispersal affects individual traits although the reversal is more obvious; community structure is obviously a phenomenon at the community not the population level of biological organization.
L37. The phrase "not a static event but differs between as well as within species" is confusing. What does 'static event' mean? Why do you imply that a difference within species is more obvious or logical than a difference between species?
L40-41. Very vague sentence.
L42. Dispersal kernel needs a more rigorous definition.
L51. If you are considering performance ability to be maximal exertion, was this maximum swimming speed or some similar measure, rather than average or some other measure unrelated to maximum exertion?
L70. Because the dispersal kernel is a statistical description of a population trait, it is not clear that it can have a function.
L78. Wordy and unclear what you mean by 'almost unnoticed occurrences'.
L80. Is Xenopus really the most widespread species in Africa? More than humans, cattle, dogs, and cats? Do you mean species of wild vertebrates or amphibians or this genus?
L100ff. The description of the study site could be improved.
• The paragraph starts with a description without specifically defining the site. I would have expected a topic sentence for this paragraph such as 'The study site was an area east of Kleinmond measuring XX meters by YY meters including AA permanent and BB temporary water bodies as well as CC intermittent streams.'
• The description should include latitude and longitude as well as the number of water bodies involved. (Note that there are discrepancies between parts of the manuscript whether there are 7 (L70) or 8 (L121).
• It seems to me that possible overland movements are constrained by the location of water bodies available. A statement indicating at least the minimum and maximum possible distances in this study would be useful.
• A figure would be appropriate. Because Fig. 2 is not mentioned until late in the Results, I suggest splitting it so that Fig. 1 includes the panels relevant to the study site and the remaining panels (becoming Fig. 3) address the findings. (Perhaps you had this in an earlier version as there is a reference to a non-existent Fig. 1.1 on L122.)
• Panel a would be aided by a larger scale map of Africa because the southwestern cape won't be recognized by some readers.
• I also find panel b challenging because some of the water bodies are too small to be easily perceived. I suggest replacing it with a map using a white background so all the water bodies are clearly visible. If you feel that the satellite image is useful, you could add a third panel to illustrate the locations.
• You name certain sites in the Results. If these locations are relevant, you should provide names on the map; otherwise, describe rather than name them in the Results.
• The map(s) of the study site should have an indication of north and a scale for distance.
• You should credit sources for the map and photos.
L103. After defining genus and species, use X. laevis consistently (rather than Xenopus laevis as you sometimes do) when referring to the study species.
L104. Why is there a reference to X. gillii, which seems to imply it is of direct relevance to the questions asked. As suggested by a reviewer, an explicit statement about all co-occurring anurans and where they occur might be clearer and more useful.
L120ff. Please clarify the sampling design. How many sampling sessions occurred and what were the intervals between them? How were sampling sessions distributed among the water bodies?
L134. The sex ratio calculation belongs in the Data Analysis section, not in the middle of your description or measuring and tagging. Also, the equation is ambiguous and needs parentheses to be added; as written, the 100 could be multiplied by the denominator. This is an unfamiliar measure of sex ratio compared to the format (male number):100 females or a percent male (or female). If it is widespread format, you can leave it; otherwise, switch to a more conventional approach. Finally, I do not see any indication in the Results that you used the calculation of sex ratio for each session as stated here. If relevant, you also need to indicate whether this was also calculated for each water body.
L135. Wouldn't you scan the frogs first to detect previous tags and only tag those without a tag? This section of the capture/recording methods needs a little reorganization for logical flow.
L139-146. I think that sex ratio, distance calculation, data transformation, and tests of normality and homogeneity belong more logically in the data analysis section. Distance could be combined with seasonal patterns (L183-184). This would be a good place to indicate that SD (appropriate for descriptive data) indicates variability throughout the manuscript.
L174-181. I think your manuscript would be easier to follow if you incorporated the analysis of differences in size and sex between single and multiple captures into your Results. I have suggested where to insert these findings in the first Results paragraph in my comments on the pdf. The related results for the sex and size of those that moved and did not move (L239-242) could be incorporated after the first sentence of the second paragraph (L211).
• It is important to give the values for size and sex for the single and multiple captures as well as the test statistics and p-value.
• You don't need 6 decimal places for chi square.
• L178 is logically incorrect. From your analysis, animals moving more than once were larger. You did not test whether larger animals moved more which would require sorting your sample by size and looking at movement for each size class.
• I suspect your analysis regarding size would be more rigorous if you separated the sexes before comparing sizes of those captured once vs. multiple times.
L185. Was the size used the size at first capture or the size when captured at the new location?
L194-197. With regard to MANCOVAs, please be more specific about what questions you were attempting to answer. Also, is 'determinate variable' the correct term? To me, it sounds like a reference to the independent rather than the dependent variables, but that may be a limitation of my statistical knowledge.
L201. Capture event was not defined.
L202. It should be clear what measure of variation you are using. Consider whether presenting the range would also be useful in many of your descriptive data summaries. This is one of those places.
L205-206. Why is total males plus females more than total individuals?
L210. Since this is probably not normally distributed, provide the median also.
L211. I was surprised that you did not compare distances moved between males and females, given your sex comparisons of probability of movement and performance. If there were any differences, would there be any benefit in calculating dispersal kernels for each sex separately as well as combined? It would seem quite easy to indicate the distances for the two sexes with only a minor modification (different shading for males and females in the bars in Fig. 1.) On L338 you state that males and females moved the same distances, but I did not notice the information supporting this statement.
L211. Why write 'only' 5 returned? This is nearly half the sample and seems quite a high proportion to me.
L217. Provide the definitions and units for X and Z.
L219. Do you really mean that only 69 of 108 movements occurred within a season? Wasn't it likely that all movements took place within a very short period (hours or days) and therefore within a season but that documentation of locations before and after the move occurred within the same season in only 69 cases?
L221. 'Between sites between seasons' is a bit ambiguous, considering that the previous statement address within-season movement, the opposite of which would be 'between seasons'. Do you mean 'of the 69 individuals for which the season of movement was known because they were recorded at both the origin and destination sites in the same season, there was no difference between seasons in either the number of animals moving (give numbers in wet and dry as well as chi square) or in the sex of animals moving (numbers of males and females and chi square)'?
L231. How is recapture of the same individuals two months later evidence for synchronous movement? Couldn't they have moved at different times over that period?
L239-242. For both sex ratio and size, comparisons of individuals that moved vs. those that did not approached statistical significance. Would it not be more appropriate to state something like 'Although 69% of individuals that moved were female, this was not quite statistically significantly different from the 60% of females in the captured population (statistical info).' This consideration might lead you to not refer so decisively to a lack of sex bias in movement (Abstract, Discussion L282). For size, 'Individuals that moved tended to be larger than individuals that were recaptured in the same water body (statistics).' [Note that I could not find information as to whether they tended to be larger or smaller. You should provide the sizes.] In addition, with a trend toward female bias in moving, it would be appropriate to separate males and females for the size comparisons.
L244. Table 2 needs to include the mean, SD and range of actual measurements because the statistics are not meaningful without the values. Also, I did not see any discussion of these data. If the information was worth collecting and presenting to readers, it is worth discussing its relevance.
L247. I don't see how female toe length is an exception to the larger forelimb measures of males because it is not a forelimb measure.
L249. Incorrect wording: I think you mean that the sexes were significantly different in time to exhaustion when size was considered. I'm not sure what 'corrected size' means.
L248ff. For distance moved and maximum duration of movement, the distance and duration of movement of each sex is as relevant to the ecological discussion as the size-corrected distance. Please provide means, SD, range and statistical comparisons for the raw distance and duration as well as for relative distances.
L253. Although Reviewer 2 suggested reducing the Discussion by 30%, I did not feel it was excessively wordy or speculative, given the topic. No reduction is needed from my perspective.
L304. How do your performance data show that males and females are equally able to disperse long distances? I don't think you presented the unscaled distances moved and related statistics. It is not clear how distance is scaled to duration; the sentence implies that males moved for a proportionately longer time, but it is not clear what this means. (Concept also appears on L309.)
L312. I think it would be useful to present your calculation for the minimum number of nights required for the longest movements. Don't you have to take into account that animals may not be moving at maximal capacity in the field and hence may move for a longer period and cover a greater distance, in addition to the potential to rest and recover during the dispersal event?
L321. If 'sub' refers to a submitted publication that still has not received final acceptance at the time of your revision, cite it as a personal communication from the authors.
L325. Your statement that more individuals moved during the wet season does not correspond to my understanding of what you mean to say on L221 of Results.
L330. It is not clear what you mean by the lack of directionality, because direction of movement was not recorded.

·

Basic reporting

There are no doubt differences between South African English and US English, but I found the following to be noteworthy enough to include:

Make a clear distinction between dispersion and migration

Pp. 61, 62: no reason to cite a reference twice in the same sentence.

p. 82: Poyton should have a date - a full citation

p. 93: delete “When it occurs …

Line 183: Why add the word “event” to movement? Not necessary, besides, “occurrence” may be a better term

Line 218: This truth of this sentence depends on capture-recapture techniques used. If using drift fence/pitfall trap arrays, this statement is false.

Line 240: Replace “to” with “from.”

Line 272: How about “burrowed animals” or “buried animals?” This phrasing sounds microbial.

Line 304: replace “further” with “farther.” Farther is the term relating to dostance.

Lines 316-317: How can sites receive movement? Animals move.

Line 324: Remove “do”

Line 333: Move this to the bottom of the paragraph – the least important conclusion.

Line 345: Delete “would like to”

Also, much of the North American literature on migration, dispersal, and population dynamics is left out. There is one citation, though, you must incorporate: Smith, M. A., and D. M. Green. 2005. Dispersal and the metapopulation paradigm in amphibian ecology and conservation: Are all amphibian populations metapopulations? Ecography 28:110–128.

You should also make a clear distinction between dispersion, migration, and perhaps habitat abandonment or some such other term.

Experimental design

Very good. I have no issues.

Validity of the findings

High, plus these findings are important.

Additional comments

Nice design, important study, clunky word usage and at least one NA citation should be added.

·

Basic reporting

No comment

Experimental design

No comment

Validity of the findings

I have some major concerns regarding on the statistical analyses used here.

Additional comments

This manuscript examines the dispersal of a native population of Xenopus laevis. Most of individuals moved less than 200 m, with a maximal distance reported of more than 2 km. No sex-biased dispersal have been identified in this study. This is an interesting paper, for which the dispersal data of this invasive species might be very useful in models.

The manuscript is well-written. I have some major concerns regarding on the introduction and the statistical analyses used here. I recommend the acceptance of this publication but after several corrections. I have detailed my major and minor comments below, and I hope that they will contribute to improving this manuscript.

Introduction

Page 6, Line 47. I suggest to change “Locomotion” to “Movement”. In my point of view, the term “locomotion” is commonly used related to the way to move (for instance S-shaped movement in salamander species), whereas the movement is linked to both the migration and dispersal processes (with dispersal kernels, dispersal propensity and distance moved).

Page 6, Line 49: Dispersal is a multi-causal process, which could be measured using two methods: either by tracking animals in the field (using capture-recapture for example) or, because dispersal is a movement which could induce gene flow, molecular tools are also used to determine the dispersal ability of a given species. Consequently, dispersal is not only estimated by the observed distance moved by individuals. Doing so, I suggest to change “whereas dispersal represents the observed distance dispersed” to “whereas dispersal represents the distance moved”. More details about dispersal are needed in this part.

Page 6, Line 50. Of course, performance abilities might be correlated to the mobility capacities in a species. I am surprised to not see the papers from Herrel and Bonneaud here. Herrel and Bonneaud (2012) and Herrel et al. (2014) demonstrated several relationships between locomotion abilities and performances in a sister species Xenopus laevis.


Herrel A, Bonneaud C (2012) Trade-offs between burst performance and maximal exertion capacity in a wild amphibian, Xenopus tropicalis. J Exp Biol 215:3106–3111

Herrel A, Vasilopoulou-Kampitsi M, Bonneaud C. (2014) Jumping performance in the highly aquatic frog, Xenopus tropicalis: sex-specific relationships between morphology and performance. PeerJ 2:e661

Page 7, Line 95. You did not define the difference between dispersal (a movement which could induce gene flow) and migration. Maybe you could add few lines about this difference (from line 49 for instance) that should help for the understanding of the paper.

Materials & Methods

Page 8, Line 113. Do you have more information about the permanent and temperature water-bodies? Temperature of water, width, depth, or if other amphibian species are living within?

Page 8, Line 121. How many tracking session? How many days between them?

Page 9, Line 128. Did you applicate a protocol to avoid the dissemination of Bd (Batrachochytrium dendrobatidis) after each tracking session?

Page 9, Line 135. Did you anesthetize individuals before pit-tagging? You should be, please details the method used.

Page 9, Line 145. Please give the p-value of the Shapiro test for testing the normality of data.

Page 9, Line 149. Please explain why you perform performance tests additionally to the capture-mark-recapture session.

Page 10, Line 166. Why did you chose the longest distance performed in the shortest time in your analyses? You can keep the 3 distances by individuals and run mixed-effect models by adding the individual as random factor in your models. By taking into account the intra-individual variation of the distance performed in controlled conditions, this statistical method would increase the robustness of your data.

Page 10, Line 179. You could give the results using the complete dataset for the tests related to the size.

Page 10, Line 184. Your analysis tacking into account both the dry and wet season is interesting. But it could be very pertinent to have the humidity conditions recorded at each capture session in order to relate these data with the movement events.

Results

Page 11, Line 203. Please, give the range of sex ratio in percent.

Page 11, Line 213. Remove “as is usual in amphibian dispersal” (which is more appropriate in the Discussion part).

Page 12, Line 225. “The maximum dispersal distance observed was a X. laevis female, which travelled 2 420 m” is a repetition of the line 214. Please add details about this long-dispersal event at line 214. Also, give the distance in km or in meters throughout the manuscript.

Discussion

Discussion could become reduced by at least 30% without losing essential information.

Page 14, Line 294. In fact, sex-biased dispersal found in amphibians have been revealed using genetics, which suggest that juveniles (individuals that we cannot tag in most of amphibian species) are responsible for most of dispersal events (inducing gene flow).

Line 341. Do you envisage performing a radio-tracking study in this native population?

---

## Round 0.2 · Minor Revisions

The manuscript has been improved by the changes. However, a few issues still remain:

L43. I had requested a definition of dispersal kernel to help readers understand the concept. However, I find your revised statement "made up from the different Euclidian distances between source and destination points for a population" to be more of a description of components than a proper definition. Can you come up with a clear definition?

L201. I suggested that it was preferred practice to use SD rather than SE as a measure of variability for descriptive data. I see that you used SE and did not respond to this point in your rebuttal. Do you have a valid argument for retaining the use of SE or can you change the variation to SD? Note that changes will also apply to Table 2.

L237-238. For the seasonal pattern of distances moved, the median is much less than the mean in the wet period and much greater than the mean in the dry period. While not mathematically impossible, this would require strong opposite skews in the distribution of distances in the two seasons. It looks to me as though the median (or mean) numbers were switched between seasons. Can you check your data to be sure the manuscript values are correct?

Need for checking the data: If these values for distances moved were, in fact, reversed, I would recommend a careful review of all numerical values in the manuscript. This suggestion is based on two errors that I detected (one in this version, one in the previous version) based only the logic. This implies that other errors that were not so obvious or that I failed to notice are present. You and I as well as PeerJ should try to be as certain as possible that there are no errors in the published values.

L255, 257. Statistical results without the data patterns are much less valuable to readers. Can you please provide means, SD or SE, ranges, and sample sizes for the relationship between sex and distance moved and between size and whether movement between water bodies occurred?

Fig. 1. The caption to this figure requires revision. The color scheme on the map presumably applies to panel (a) as well as panel (b) but the caption implies that it only applies to (b). Did you obtain the basic map from a source that should be credited?

My editorial decision noted that your submission lacked Supplementary Information for the non-movers and for the laboratory study. Your rebuttal did not address this point, and the Supplementary Information is still lacking.

Finally, the manuscript still contains a few errors in grammar, spelling, and punctuation. I have attached a pdf with highlights to indicate the problems I noticed.

---

## Round 0.3 · accepted · Accept

The manuscript is now suitable for publication.